# Regulative Loop between β-catenin and Protein Tyrosine Receptor Type γ in Chronic Myeloid Leukemia

**DOI:** 10.3390/ijms21072298

**Published:** 2020-03-26

**Authors:** Luisa Tomasello, Marzia Vezzalini, Christian Boni, Massimiliano Bonifacio, Luigi Scaffidi, Mohamed Yassin, Nader Al-Dewik, Paul Takam Kamga, Mauro Krampera, Claudio Sorio

**Affiliations:** 1Department of Medicine, General Pathology Division, University of Verona, 37134 Verona, Italy; 2Department of Cancer Biology and Genetics, The Ohio State University, Columbus, OH 43210, USA; 3Comprehensive Cancer Center, The Ohio State University, Columbus, OH 43210, USA; 4Department of Medicine, Section of Hematology, University of Verona, 37134 Verona, Italy; 5National Center for Cancer Care and Research, Department of Hematology and BMT, Hamad Medical Corporation, Doha, Qatar and College of Medicine Qatar University, Doha, Qatar; 6Pediatrics department, Women’s Wellness and Research Center (WWRC), Interim Translational Research Institute (iTRI), Hamad Medical Corporation (HMC) and College of Health and Life Science (CHLS), Hamad Bin Khalifa University (HBKU), Doha, Qatar; 7Stem Cell Research Laboratory, Section of Hematology, Department of Medicine, University of Verona, 37134 Verona, Italy

**Keywords:** β-catenin, PTPRG, chronic myeloid leukemia, methylation, tumor suppressor, tyrosine phosphatase, DNMT, 5-azacytidine

## Abstract

Protein tyrosine phosphatase receptor type γ (PTPRG) is a tumor suppressor gene, down-regulated in Chronic Myeloid Leukemia (CML) cells by the hypermethylation of its promoter region. β-catenin (CTNNB1) is a critical regulator of Leukemic Stem Cells (LSC) maintenance and CML proliferation. This study aims to demonstrate the antagonistic regulation between β-catenin and PTPRG in CML cells. The specific inhibition of PTPRG increases the activation state of BCR-ABL1 and modulates the expression of the BCR-ABL1- downstream gene β-Catenin. PTPRG was found to be capable of dephosphorylating β-catenin, eventually causing its cytosolic destabilization and degradation in cells expressing PTPRG. Furthermore, we demonstrated that the increased expression of β-catenin in PTPRG-negative CML cell lines correlates with DNA (cytosine-5)-methyl transferase 1 (DNMT1) over-expression, which is responsible for *PTPRG* promoter hypermethylation, while its inhibition or down-regulation correlates with PTPRG re-expression. We finally confirmed the role of PTPRG in regulating BCR-ABL1 and β-catenin phosphorylation in primary human CML samples. We describe here, for the first time, the existence of a regulative loop occurring between PTPRG and β-catenin, whose reciprocal imbalance affects the proliferation kinetics of CML cells.

## 1. Introduction

Chronic myeloid leukemia (CML) is a myeloproliferative disease affecting approximately 1 per 200,000 persons per year in industrialized countries. Many treatment improvements have been achieved recently, especially in the development of new drugs, but a mortality rate of 2%–3% per year remains [1]. A distinctive feature of CML is the reciprocal translocation, originating in hematopoietic stem cells (HSCs), between the long arms of chromosomes 9 and 22, i.e., t (9;22) (q34;q11). This genomic aberration generates a new fusion gene, BCR-ABL1, which encodes for a tyrosine kinase held accountable for the neoplastic transformation of these cells by affecting normal cellular pathways essential for tissue homeostasis, and thus causing the alteration of crucial cellular processes, such as apoptosis, cell cycle and autophagy [2,3].

In this context, one primary goal of the research is to identify the regulatory mechanisms antagonizing the kinase activity of BCR-ABL1 and, possibly, of other vital effectors intersecting this pathway, as players other than BCR-ABL1 have been involved in the pathogenesis of the disease [4]. Protein tyrosine phosphatase receptor type γ (PTPRG) is a member of the protein tyrosine phosphatase (PTP) family, featuring an extracellular and a single transmembrane region and two tandem intracytoplasmic catalytic domains [5]. PTPRG is widely expressed in human tissues [6] and is involved in the regulation of cell growth, differentiation, mitotic cycle, and oncogenic transformation [7,8]. The gene encoding for this phosphatase is located in a chromosomal region (3p21-p14), frequently deleted in renal cell and lung carcinoma, where PTPRG acts as a tumor suppressor [9,10,11]. Epigenetic events, such as hyper-methylation of its promoter region, negatively regulates the transcription of PTPRG, as demonstrated in CML and childhood acute lymphoblastic leukemia [12,13] Re-expression of this protein occurs in the leukocytes of CML patients following targeted therapy [12]. Once activated, PTPRG can reduce the phosphorylation level of BCR-ABL1 and some of its key targets, such as CRK-L and STAT5 [12]. β-catenin (CTNNB1), the most critical nuclear effector of Wnt signaling, is an essential component of cadherin-based adherents junctions. The cellular levels of β-catenin are regulated by its phosphorylation and consequent binding to the destruction complex, including APC, Axin1, glycogen synthase kinase-3 (GSK3β) and casein kinase I (CKI) [13,14,15]. Once stabilized, β-catenin moves to the nucleus and binds TCF4/LEF (T-cell factor/lymphoid enhancer factor) transcription factors, thereby promoting the transcription of genes involved in neoplastic transformation, such as MYC, Cyclin D1 (CCND1) and the DNA (cytosine-5-)-methyltransferase 1 (DNMT1) [16]. This methyltransferase, with a preference for hemimethylated sites, causes the down-regulation of different tumor suppressor genes [17]. BCR-ABL1 phosphorylates β-catenin on Tyrosine 654, thus increasing its stability and nuclear translocation. Imatinib mesylate (IM), a specific inhibitor of BCR-ABL1 activity, causes a decrement of β-catenin expression, both in established cell lines and in primary cells derived from CML patients in blast crisis [18]. We found that, in CML cells, higher expression of PTPRG correlates with dephosphorylation and increased β-catenin-degradation. Moreover, we demonstrated that they are part of the same molecular complex. At the same time, β-catenin promotes DNMT1 transcription, causing PTPRG silencing through the hypermethylation of its promoter region, thus indicating a pivotal role in the BCR-ABL1 activated pathway.

## 2. Results

### 2.1. PTPRG Regulates the Expression of β-catenin through Its Tyrosine Dephosphorylation

K562 cells were stably transfected with PTPRG cDNA (K562 γ1), to restore the expression of PTPRG in this CML cell line (Figure 1A). We also subcloned LAMA-84 (another CML cell line) cells and isolated high and low PTPRG-expressing clones, as demonstrated by qRT-PCR and immunoprecipitation experiments (Figure 1B). We analyzed the expression of total and tyrosine-phosphorylated BCR-ABL1 and β-catenin proteins in K562 cells expressing or not PTPRG (Figure 1C) and in both LAMA-84 clones (Figure 1D), after the treatment with a PTPRG inhibitor (PTPRG IN) [19] at a concentration of 2 and 10 µM. We demonstrated that, in PTPRG-positive cells, the inhibition of PTPRG leads to the BCR-ABL1 Y245 and β-catenin Y654 phosphorylation increment, in tandem with a raised level of total protein (Figure 1C,D,F,G). Opposingly, PTPRG-negative cells were not responding to the treatment with PTPRG IN, in terms of the effect on phosphorylation or total expression levels for these two proteins: they remain highly-expressed at levels comparable to K562 γ1 after the inhibition of PTPRG. We confirmed that inhibition of BCR-ABL1 activity following treatment with imatinib mesylate in K562 cells causes the dephosphorylation of β-catenin Y654 and consequent protein degradation, as demonstrated by Gambacorti-Passerini’s group in 2008 [18]. The interference with BCR-ABL1 activity by PTPRG is further supported by the finding that, in K562 γ1 cells, β-catenin and its Y654 phosphorylated form are strongly reduced in comparison with the mock control clone, making the evaluation of relative phosphorylation level challenging to assess (Figure 1E). According to the data described above, the knock-down of PTPRG through RNA interference in unfractionated LAMA-84 cells (Figure 1H) resulted in an evident increase of phospho- and total- β-Catenin, as demonstrated by Western blotting and immunofluorescence experiments (Figure 1I,J). 

In summary, in PTPRG-positive leukemic cells, the activity of this phosphatase causes the decrease of BCR-ABL1 and β-catenin tyrosine phosphorylation and the resulting degradation of the latter. 

### 2.2. β-catenin and PTPRG Belong to the Same Protein Complex

Previous studies reported that BCR-ABL1 increases β-catenin stability through its phosphorylation activity in CML cells [18]. It can thus reasonably be expected that BCR-ABL1, β-catenin, and PTPRG belong to the same protein complex. So, we first evaluated the presence of β-catenin and PTPRG in a multi-molecular complex, by performing pull-down assay experiments. As a “bait” protein, we used the inactivated PTPRG substrate-trapping mutant (D1028A) [12] recombinant protein immobilized on nickel-agarose beads and challenged with protein extracts from K562 (BCR-ABL1 positive) and U937 (BCR-ABL1 negative) cell lines. This approach detects trapping of the substrate into the PTP catalytic pocket [20] and has allowed us to evaluate whether an interaction between PTPRG and β-catenin occurs. Co-precipitation happened in both cell lines, suggesting that this interaction does not require the expression of the oncogene BCR-ABL1 (Figure 2A). Co-immunoprecipitation experiments in cells *in vivo* expressing the substrate trapping form of PTPRG (K562 D1028A) confirm that β-catenin also binds PTPRG in a fully native condition (Figure 2B). 

### 2.3. PTPRG Expression Increases β-catenin Affinity Binding to Its Degradation Complex

The loss of β-catenin Y654 phosphorylation by BCR-ABL1 leads to the binding between β-catenin, and its multi-protein “destruction complex”, that includes the tumor suppressors Axin1 [21] Axin1 carries binding sites for CK1 and GSK-3β and coordinates the whole phosphorylation events, involving β-catenin proteolysis by the 26S degradation machinery and acts as a tumor suppressor in hepatocellular carcinoma [21,22]. Immunoprecipitation experiments determined that, in K562 cells, β-catenin is present in a complex with Axin1 only in the presence of PTPRG (Figure 2C). Besides, the restored expression of PTPRG in the K562 cell line (K562 γ1) induces a strong up-regulation of Axin1, along with the down-regulation of β-catenin, compared to PTPRG-negative K562 (K562 mock) (Figure 2D).

As the additional confirmation of β-catenin-proteolysis events driven by PTPRG, we inhibited β-catenin degradation using the proteasome 26S subunit inhibitor, MG-132, in K562 cells expressing, or not, PTPRG. The proteasome inhibitor blocked the degradation of the protein, but dephosphorylation still occurred in the presence of active PTPRG (Figure 2E), indeed suggesting that PTPRG dephosphorylates β-catenin. To further strengthen this finding, we treated the unfractionated PTPRG-positive LAMA-84 for two hours with the proteasome inhibitor MG-132 (10 µM) alone, and with the two inhibitors PTPRG IN (10 µM) and MG-132 (10 µM) combined, respectively. In keeping with the proposed mechanism, restored expression of the total β-catenin protein was associated with an increased level of Y654 phosphorylation in the sample treated with both inhibitors. Conversely, the blocking of the 26S proteasome without the inhibition of the phosphatase PTPRG prevents the degradation of the total protein but does not increase its phosphorylation level (Figure 2F). In other words, the lowering of Y654 phosphorylation levels is not the result of β-catenin degradation, but of PTPRG activity driving β-catenin degradation.

### 2.4. PTPRG Down-Regulates β-catenin and Affects the Expression of Its Transcriptional Targets

We next evaluated whether β-catenin transcriptional targets were vicariously affected by PTPRG expression in leukemic cells, as should be expected by the previous findings.

TCF4/β-catenin complex is known to inhibit the transcription of the *p21/WAF1* gene [23]. In keeping with this observation, we noticed that the higher expression of PTPRG is closely related to the reduced expression of β-catenin and consequent overexpression of *p21/WAF1* in K562 cells (Figure 3A), while the contrary was observed when PTPRG is down-regulated by a specific siRNA (siPTPRG) in LAMA-84 cells (Figure 3B). Similarly, the up-regulation of PTPRG is also associated with the decreased transcription of two genes positively regulated by β-catenin and involved in cellular proliferation: *MYC* [24], and *β-catenin* itself, which acts as a co-transcription factor (with TCF4/LEF) on its promoter [25] (Figure 3A). Opposite results were obtained when PTPRG is down-regulated in the PTPRG positive CML cell line (LAMA-84), further confirming the finding (Figure 3B).

Differently from what was expected, the regulation of the *Cyclin D1* gene, positively regulated by β-catenin, in our experimental conditions was not inversely related to PTPRG expression. The boost of Cyclin D1 expression is likely to be offset by the up-regulation of p21/WAF1, both at mRNA and protein levels (Figure 3A–C), thus avoiding an unchecked activation of the cell cycle. Moreover, we demonstrated by Western blotting that, despite the fact that PTPRG increases the expression of *Cyclin D1* mRNA, the protein is mostly localized in the cytosolic compartment in PTPRG positive K562 cells (Figure 3C), thus preventing its activity as a cell cycle inducer in the nucleus.

Given these premises, we anticipate that PTPRG inactivation by PTPRG IN should have a positive impact on CML cell proliferation. To test this hypothesis, we performed dose-response experiments and evaluated the extent of BCR-ABL1 protein expression and Y245 phosphorylation. As expected, there is a dose-dependent increase of colony size and volume (Figure 3D,E), that correlates with the enhancement of Y245 phosphorylation (Figure 3F), thus confirming and extending our previous observation on PTPRG involvement in the regulation of CML cell proliferation [12]. Interestingly, the samples that showed the highest number and size of colonies (treatment with 0.1 and 0.2 µM PTPRG IN) were the same ones that expressed the highest level of phospho-BCR-ABL1, as demonstrated by Western blotting (Figure 3F).

### 2.5. β-catenin Transcriptional Activity Correlates with DNMT1 Expression

Since we have previously demonstrated that PTPRG expression is linked to the promoter methylation levels [12], we hypothesize that DNMT proteins might negatively affect the transcription of this phosphatase. We focused on DNA (cytosine-5)-methyltransferase 1 (DNMT1), a DNA-binding enzyme responsible for the down-regulation of many tumor suppressor genes through hypermethylation of their promoter regions and a downstream effector of APC/β-catenin/TCF4 signaling [26]. It has been reported that the inhibition of β-catenin/TCF4 transcriptional activity, through the N-terminal deletion dominant-negative mutant, decreases *DNMT1* mRNA levels [17]. Interestingly, *DNMT1* transcript inversely correlates with PTPRG expression, as shown by the differential expression level in K562, LAMA84, and PTPRG-silenced LAMA84 (Figure 4A,B). Moreover, in K562 cells expressing β-catenin (K562 mock), the β-catenin signal disruptor PNU-74654, which prevents β-catenin binding to TCF4/LEF transcriptional co-factor, leads to the down-regulation of *DNMT1* mRNA (Figure 4C).

Three active DNA methyltransferases have been identified in mammals: DNMT1, DNMT3a, and DNMT3b. DNMT1 is considered essential for the maintenance of DNA methylation, while DNMT3a and DNMT3b are involved in the first step of methylation [26]. We detected, by Western blotting, the higher expression of DNMT1 and DNMT3b in PTPRG-negative K562 cells compared to PTPRG expressing LAMA-84 cells, while DNMT3a, considered as a tumor suppressor in hematological malignancies [27], followed an opposite trend of expression, being present only in LAMA-84 cells (Figure 4D).

### 2.6. DNMT1 Binds PTPRG Promoter Region and Regulates PTPRG Expression

We therefore evaluated whether DNMT1 could represent a modulator of PTPRG expression through the regulation of its promoter methylation status. As demonstrated by previous works [28,29], DNMT1 can bind gene promoter regions to control their transcription by methylation. Furthermore, the *PTPRG* promoter region contains two distinct CpG islands available for methylation by DNMT1 and, consequently, potential binding sites for this protein. 

Chromatin immunoprecipitation assay is a powerful technique suitable to map the interaction of proteins with specific genomic regions. This assay confirmed the interaction between DNMT1 and the CpG islands within the *PTPRG* promoter region (Figure 4E), indicative of a role for DNMT1 in PTPRG down-regulation.

As further confirmation of our data, we sequenced the genomic region of the *PTPRG* promoter, after bisulfite conversion of the DNA from K562 and LAMA-84 cells. The sequencing analysis revealed a solid methylation pattern of this region only in cells with low expression of PTPRG (Figure 4F,G), stronger in K562 cells compared with both clones (high and low-PTPRG expression) of LAMA-84.

In order to validate the involvement of DNMT1 in PTPRG transcriptional regulation, we treated PTPRG-negative K562 cells with a specific siRNA against DNMT1, alone or in combination with 5-azacytidine, an inhibitor of DNA methyltransferases (Figure 4H). Although even a single treatment leads to a restored expression of *PTPRG*, the combination of 5-azacytidine with DNMT1 siRNA caused a 5-fold increment of *PTPRG* mRNA in comparison with control. Increased mRNA expression matches with the recovery of detectable protein levels (Figure 4I,J).

### 2.7. PTPRG Dephosphorylates BCR-ABL1 and β-catenin in CML Primary Cells

We finally tested whether PTPRG affects BCR-ABL1 and β-catenin phosphorylation in primary CML cells. We first evaluated by qRT-PCR (data not shown), and flow cytometry the level of PTPRG in peripheral blood and bone marrow leukocytes obtained at the time of diagnosis from nine CML patients (Table 1) (Figure 5A,B left panels). Five CML patients expressed detectable PTPRG levels, while four were PTPRG-negative. We tested the phosphorylation level of BCR-ABL1 in all nine patients and that related to phospho β-catenin in 6 out of 9 patients. In all the patients expressing PTPRG, 10µM PTPRG IN enhanced Y245 phosphorylation of BCR-ABL1 and, consistently with our previous results, Y654 phosphorylation of β-catenin (Figure 5A,B right panel). Interestingly, two patients samples presenting undetectable levels of PTPRG (MFI 1.15, patient 3 and 1.03, patient 8) did not show increased BCR-ABL1 Y245 phosphorylation in response to PTPRG IN (Figure 5A–D). These results fully confirm and extend to primary cells the previous data that indicate an active role of this phosphatase in BCR-ABL1 activation and β-catenin degradation and, consequently, the inverse correlation between PTPRG expression and BCR-ABL1 activity in cells isolated by the peripheral blood of CML patients.

## 3. Discussion

Our previous works demonstrated the role of tumor suppressor for PTPRG in CML [12] and, in a different cell context, its capability to target JAK2 [30], a kinase governing pathways of relevant therapeutic interest in CML [31]. Here, we extend these previous findings by exploring in deeper detail the molecular mechanisms underlying the tumor-suppressive effect of this phosphatase. We proved that PTPRG dephosphorylates the residue Y245 in the BCR-ABL1 SH2–kinase-domain linker, essential for the full activation of this kinase and one of five high-confidence phosphorylation sites of BCR-ABL1, with structural and functional roles [32]. Dephosphorylation, with consequent BCR-ABL1 inactivation or weakening, represents a dramatic event for CML cells leading to increased apoptosis and differentiation, followed by reduced proliferation and aggressiveness [33]. This evidence is of considerable significance, thinking the role of Leukemic Stem Cells (LSC) in the resistance to Tyrosine Kinase Inhibitors and aberrant activation of self-renewal [34]. In the last few years, many studies have suggested the active contribution of β-catenin on LSC maintenance [35], even if the relationship between β-catenin and BCR-ABL1 is still under investigation. Some authors described BCR-ABL1 as a β-catenin target [36], demonstrating that β-catenin-deficient murine CML cells showed lower BCR-ABL1 protein level and phosphorylation activity. MYC, a β-catenin-downstream gene, controls BCR-ABL1 transcription, suggesting that BCR-ABL1 is an indirect target of β-catenin [37]. Furthermore, lncRNA-BGL3 was highly induced in response to the disruption of BCR-ABL1 expression, or by inhibiting BCR-ABL1 kinase activity in K562 cells and leukemic cells derived from CML patients. Conversely, BCR-ABL1 represses lncRNA-BGL3 expression through MYC-dependent DNA methylation [38]. Moreover, BCR-ABL1 regulates β-catenin through the phosphorylation of Tyr 86 and 654 that affects the stabilization of cytosolic β-catenin and its binding with its degradation complex [18]. Altogether, these data show that β-catenin, as Wnt pathway effector, is required to maintain normal HSC function, while the loss of this protein hampers CML progression [36]. Of note, nuclear β-catenin is involved in intrinsic BCR-ABL1 kinase-independent TKI resistance in primary CML cells [39] and PTPRG co-precipitates with β-catenin in enterocytes derived from transgenic mice, featuring the loss of the wild-type Apc allele (ApcMin−/−). In tumors derived from these mice, reduced levels of tyrosine-phosphorylated β-catenin have been reported [40].

Collectively, our data demonstrate that β-catenin degradation is controlled by PTPRG expression: PTPRG can dephosphorylate BCR-ABL1, thus preventing β-catenin tyrosine phosphorylation and, at the same time, apparently directly dephosphorylates β-catenin, causing the almost complete proteolysis within two hours (Figure 6, left panel). Based on these experiments, we cannot assess whether the dephosphorylation of β-catenin is a consequence of BCR-ABL1 inhibition mediated by PTPRG, or the effect of a direct dephosphorylation. However, we have shown that PTPRG and β-catenin belong to the same complex, as they co-immunoprecipitate. This phenomenon is independent of BCR-ABL1 expression, as it occurs also in the BCR-ABL1-negative U937 cell line, strongly suggesting a direct interaction between PTPRG and β-catenin.

The loss of β-catenin reduces its transcriptional activity, as reflected by the down-regulation of *MYC* and *β-catenin* mRNA, in combination with *p21/WAF1* mRNA up-regulation, which may explain lower cellular proliferation rate in the presence of PTPRG.

DNA methylation is a significant epigenetic modification in mammals, and, under physiological condition, 80% of the whole genome, not associated with promoter sequences, is methylated. On the contrary, in cancer cells, CpG-rich promoter methylation catalyzed by DNA methyl-transferases is a common mechanism of tumor suppressor silencing [41]. In our previous work, we demonstrated that the *PTPRG* promoter is methylated and that 55% of leukocytes from CML patients treated with TKI showed demethylation of the region and a re-expression of PTPRG [12].

DNMT1 is the major DNA-methyl-transferase responsible for the maintenance of DNA methylation after DNA replication; by contrast, DNMT3a and DNMT3b are *de novo* methyl-transferases that are often dysregulated in many types of tumors, such as colon, breast and pancreatic cancers, thus representing a therapeutic target [26]. DNMT3a has recently emerged as one of the most important tumor suppressors in hematological malignancies, in particular in AML cells, featuring a high number of mutations that are involved in resistance to some types of drugs [27].

DNMT1 and DNMT3b cooperate for tumor suppressor silencing [42], and we observed a higher expression of these two proteins in K562 cells, as compared to PTPRG-expressing LAMA-84. Besides, β-catenin/TCF4 transcriptional complex regulates DNMT1 expression. Indeed, it was demonstrated that the inactive dominant-negative version of this protein complex decreases *DNMT1* expression [17].

We present here a cross-talk between DNMT1 expression and *PTPRG* promoter methylation, which was confirmed by the results of chromatin immunoprecipitation and by the evidence of increased PTPRG transcription in K562 following DNMT1 down-regulation.

These results reveal a previously unrecognized regulative loop between β-catenin and PTPRG, with β-catenin controlling PTPRG transcription. At the same time, through DNMT1, β-catenin controls both BCR-ABL1 (through MYC expression) and PTPRG itself. On the other hand, PTPRG and BCR-ABL1 regulate β-catenin levels by modulating, in opposite ways, its tyrosine phosphorylation levels. Also, PTPRG increases the expression of Axin1, a key regulator of β-catenin degradation [15].

Of relevance is the observation that the same mechanism described in CML cell lines is active in leukocytes from CML patients, where PTPRG inhibition results in the up-regulation of Y245-phospho-BCR-ABL1 and Y654-phospho-β-catenin. Conversely, cells from CML patients expressing undetectable levels of PTPRG displayed the highest levels of Y245-phospho-BCR-ABL1 and Y654-phospho-β-catenin. Of note is the fact that they were refractory to PTPRG IN treatment, a key observation further confirming an exquisite specificity of this chemical inhibitor, utilized for the first time in cellular models.

Altogether this study describes a novel PTPRG-dependent regulative loop involving critical regulatory components of the BCR-ABL1 signaling pathway (Figure 6), further strengthening the role of PTPRG as a crucial tumor suppressor in CML cells and a possible candidate for specific therapies aimed at its re-expression or activation.

## 4. Materials and Methods

### 4.1. Cell Lines and Transfection

The CML cell lines K562 (American Type Culture Collection, Manassas, VA, USA), LAMA-84 and the human histiocytic lymphoma cell line U937 (Leibniz Institute DSMZ-German Collection of Microorganisms and Cell Cultures GmbH, Braunschweig, Germany) were cultured in RPMI 1640 medium, supplemented with 10% Fetal Bovine Serum (FBS) and 2 mmol/L of Ultraglutamine 1-(Lonza, Milan, Italy). Full-length human PTPRG cDNA, mutant PTPRG D1028A, or empty vector (pCR 3.1, Thermo Fisher Scientific, Monza, Italy), used to transfect K562 cells, were previously described [12].

### 4.2. Inhibitors

K562 and LAMA-84 cell lines were treated with the PTPRG inhibitor (from now on named PTPRG IN) 3-(3,4-dichlorobenzylthio) thiophene-2-carboxylic acid [19] (BIONET sold by Key Organics Ltd, Camelford, Cornwall, UK) at 0.05 to 50 µM and with imatinib mesylate (IM) (Selleckem, Houston, TX, USA), 1 or 5 µM, for 2 h at 37 °C. The proteasome 26S unit inhibitor, MG-132, (Santa Cruz Biotechnology Inc, Heidelberg Germany) was used at 10 µM for 2 h at 37 °C. Moreover, 5-azacydine (MilliporeSigma, Milan, Italy) was used at 10 µM for 72 h at 37 °C. PNU-74654 (MilliporeSigma, Milan, Italy) was used at 20 µM for 12 h at 37 °C.

### 4.3. CML Patients

CML cells were collected from patients with untreated, chronic-phase disease, after informed consent (described in Table 1). 20 × 10^6^ cells cryopreserved in 500 µL of 90% FBS and 10% DMSO were thawed using 4 µL of deoxyribonuclease I (260 U/µL), seeded in IMDM (Thermo Fisher Scientific, Monza, Italy) with 20% FBS for three hours and then treated with PTPRG IN, at different concentrations for two hours at 37 °C. Cells were lysed in boiling “sample buffer” (SB): 160 mM Tris, 20% glycerol, 5% β-mercaptoethanol, 4% SDS, 0.01% bromophenol blue and subjected to Western blotting. The study was approved by the Ethical Committee of Azienda Ospedaliera Universitaria Integrata Verona, protocol N. 1828, 12 May 2010, ‘‘Institution of cell and tissue collection for biomedical research in Onco-Hematology”. The study was approved by the Local Ethics Committee, AOUI Verona (Permit Number: 25066); informed consent, in accordance with the declaration of Helsinki, was obtained from each patient. 

### 4.4. Clonogenic Assay

LAMA-84 and K562 cells were resuspended in IMDM (Thermo Fisher), with 2% FBS and 0.3% NaHCO_3_ (MilliporeSigma, Milan, Italy), at a concentration of 16.5 × 10^3^ or 33 × 10^3^ cells/mL. Then, 50 µL of this cellular dilution were mixed with Methocult (StemCell Technologies Germany GmbH, Cologne, Germany) and PTPRG IN at different concentrations (0.005–0.1–0.2 µM) or DMSO and transferred in 24-well plates. After 8 days, cell colonies in each well were stained with 3-[4,5-dimethylthiazole-2-yl]-2,5-diphenyltetrazolium bromide (MTT) (MilliporeSigma, Milan, Italy) and, after 3 h at 37 °C, captured and quantified with ImageQuant® LAS 4000 colony counting module (GE Healthcare Europe GmbH, Milan, Italy).

### 4.5. Small Interfering RNA (siRNA) Transfection

Small interfering RNA (siRNA) targeting PTPRG (siPTPRG, No s-11550), siRNA targeting DNMT1 (siPTPRG, No s-4215), and negative control (scrambled sequence) were purchased from Applied Biosystems^®^, Thermo Fisher Scientific, Milan, Italy. PTPRG-negative K562 and LAMA-84 cells were transfected at a concentration of 1 × 10^5^/mL with 30 nM of siRNAs, using siPORT™ NeoFX™ Transfection Agent (Applied Biosystems^®^, Thermo Fisher Scientific, Milan, Italy), according to the manufacturer’s instructions. Cells were cultured for 72 or 96 h, washed twice with cold TBS, and lysed.

### 4.6. RNA Isolation, Reverse Transcription, qRT–PCR

Cells (5 × 10^6^) were resuspended in 1 mL of TRIzol^®^ reagent (Thermo Fisher Scientific, Milan, Italy) and stored at −80 °C until the RNA extraction. Total RNA was isolated following the manufacturer’s instructions and measured with the Nanodrop 2000c instrument (Thermo Fisher Scientific, Milan, Italy).

For cDNA synthesis, 2 µg of total RNA were reverse-transcribed using random hexamer primers, with the High-Capacity cDNA Reverse Transcription Kit (Thermo Fisher Scientific, Milan, Italy) 10 min 25 °C, 120 min 37 °C, 5 min 85 °C. qRT-PCR was performed using SYBR^®^ Green PCR Master Mix (Thermo Fisher Scientific, Milan, Italy), 2 min 50 °C, 2 min 95 °C, 15 s 95 °C, 1 min 60 °C and specific primers for each gene were designed with the free, open-source GUI application PerlPrimer v1.1.21, http://perlprimer.sourceforge.net/ (Appendix A). The data were normalized with β-Actin as internal control and expressed as fold of increase of fluorescence.

### 4.7. Immunoprecipitation

Specific antibodies against β-catenin (3 µg, Santa Cruz Biotechnology Inc, Heidelberg Germany) and PTPRG [43] were added to 30 µL of Protein G-Sepharose beads (GE Healthcare Europe GmbH, Milan, Italy) and incubated for one hour at 4°C with gentle rocking. Then, washed beads bound to the antibody were added to 500 µg of protein lysates and incubated again for three hours at 4 °C with gentle rocking. Finally, the beads were washed, resuspended in Sample Buffer 2×, denaturated at 95 °C and subject to Western blotting.

### 4.8. Immunofluorescence

Cells from the LAMA-84 cell line (1 × 10^5^) treated with PTPRG siRNA or scramble sequence, were subject to cytospin (Thermo Fisher Scientific, Milan, Italy) and immediately fixed in 4% paraformaldehyde for 10 min at 4 °C. After three washing steps, slides were incubated overnight at 4 °C with primary antibodies: 5 µg/mL phospho-β-catenin (Tyr 654) (ECM Biosciences, Versailles, KY, USA), 5 µg/mL total- β-catenin (Santa Cruz Biotechnology Inc, Heidelberg Germany) and 5 µg/mL anti-PTPRG [44], in a buffer containing PBS/Tween20 0.05%/BSA 1%/NaN_3_ 4mM, followed by the proper secondary Alexa Fluor-conjugated antibodies (1:2000, Thermo Fisher Scientific, Milan, Italy), for 1 h at room temperature. After mounting with an anti-fading solution, samples were analyzed by fluorescence microscopy (DM6000B, Leica Microsystems, Wetzlar, Germay) and confocal microscopy (TCS-SP5, Leica Microsystem, Wetzlar, Germay).

### 4.9. Pull-Down Assay

Intracellular domain (ICD) of PTPRG D1028A (PTPRG inactive mutant) and enhanced green fluorescent protein (eGFP) construct, both cloned in T7-based HisG-tagged vector expression pRSET A (Thermo Fisher Scientific, Milan, Italy), were expressed in BL21 (DE3) pLysS Escherichia coli (Thermo Fisher Scientific, Milan, Italy). For the pull-down assay, 3 mg of total bacterial protein lysates were conjugated to 30 μL of HIS-Select Nickel Affinity Gel (MilliporeSigma, Milan, Italy) and incubated, after extensive washing, with 500 μg of protein lysates from K562 and U937 cell lines for 3 h at 4 °C, with gentle rocking. The beads were collected, washed twice with lysis buffer and once with cold TBS, and then subjected to SDS-PAGE and Western blotting.

### 4.10. Methylation Analysis

The DNA extraction was performed using DNeasy Blood & Tissue Kit (Qiagen Sciences Inc, Germantown, MD, USA), according to the manufacturer’s instruction. Subsequently, the DNA (2 µg) was subjected to bisulfite conversion and purification using EpiTect™ Fast DNA Bisulfite Kit (Qiagen Sciences Inc, Germantown, MD, USA), according to the manufacturer’s protocol. After bisulfite conversion, the *PTPRG* promoter region was analyzed by PCR amplification and sequencing. PCR reaction was made using 50× Advantage^®^ 2 Polymerase Mix (Takara Bio USA Inc, Mountain View, CA, USA). The PCR product was purified with ExoSAP-IT™ PCR Product Cleanup Reagent (Thermo Fisher Scientific, Waltham, MA, USA), and subjected to Sanger sequencing. The analysis of sequencing was performed with Sequencer 5.4.6 software (Gene Codes Corporation, Ann Arbor, MI, USA) to evaluate the C to T conversion rate. 

### 4.11. Western Blotting

Cells were lysed in Lysis buffer (50 mM Tris HCl pH 7.5, 150 mM NaCl, 10% Glycerol and 0,5% Nonidet P40), or in boiling Sample Buffer. Then, 10–30 µg of proteins were loaded onto 6 to 12% polyacrylamide, 0.1% SDS gels, and electro-blotted on nitrocellulose membranes (GE Healthcare Europe GmbH, Milan, Italy). Blocking solution was obtained using TBS-0.05% Tween^®^20/BSA 5% or fat-free milk 5%. All antibodies (Appendix A) were diluted in TBS-0.05% Tween^®^20/BSA or fat-free milk 1%–5%. The membranes were washed and assayed with ECL (MilliporeSigma, Milan, Italy) after overnight incubation at 4 °C, three times washing with TBS-0.05% Tween^®^20 (TBS-T) and incubation with appropriate HRP-conjugated secondary antibodies. All images were acquired with the digital imaging system ImageQuant® LAS 4000 biomolecular imager (GE Healthcare Europe GmbH, Milan, Italy).

### 4.12. Statistical Analysis

The data analysis was performed using GraphPad^®^ 8.3.0 Instat software (GraphPad Software, San Diego, CA, USA). The Student’s two-tailed unpaired *t*-test was applied to qRT-PCR and Colony assay (number and volume of colonies) experiments. For qRT-PCR analysis, we considered absolute values, even though the graphs show just the fold-change, representing the data more clearly. Also, the student’s two-tailed unpaired t-test was applied to the densitometric comparisons depicted in Figure 1F,G, Figure 5C,D. Each PTPRG IN condition was compared to the DMSO control. The analysis was done separately for each different protein employing ImageJ software (U. S. National Institutes of Health, Bethesda, MD, USA https://imagej.nih.gov/ij/, 1997–2018). Results with a *p*-value < 0.05 were considered statistically significant.

## Figures and Tables

**Figure 1 ijms-21-02298-f001:**
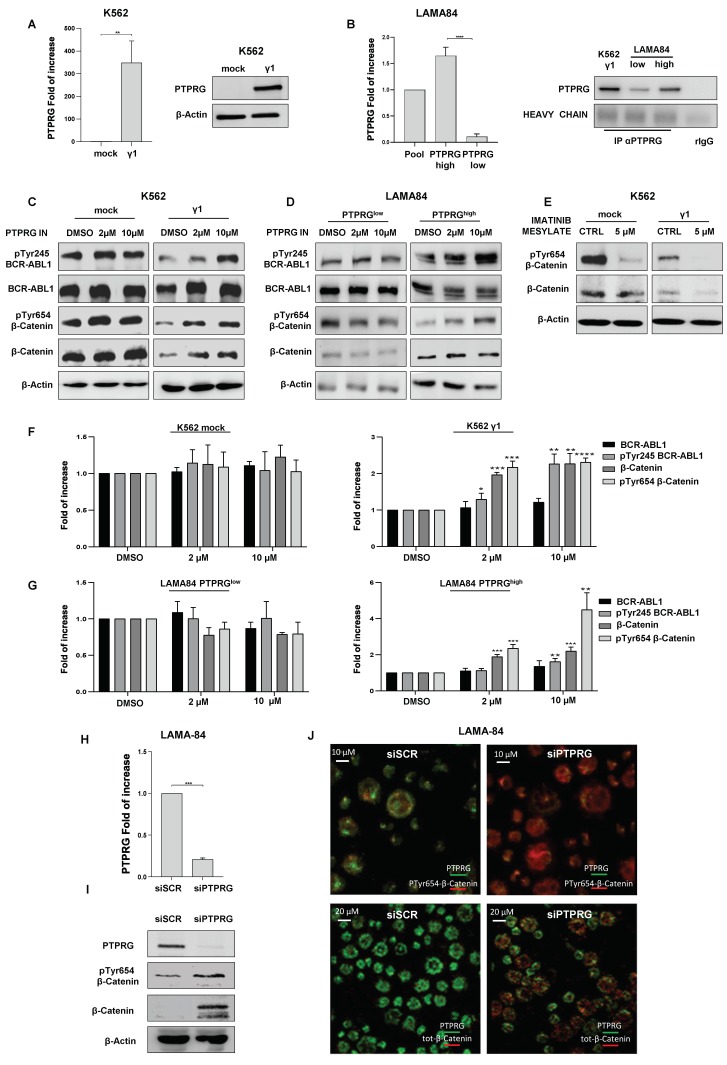
β-catenin expression and phosphorylation in K562 and LAMA-84 cell lines. (**A**) K562 cells were stably transfected with an empty vector pCR 3.1 (K562 mock) or with pCR 3.1 Protein tyrosine phosphatase receptor type γ (PTPRG) (K562 γ1) vector, containing the sequence of full-length PTPRG (PTPRG FL). qRT-PCR and Western Blotting show the fold change expression of PTPRG relative to K562 mock in transfected K562 cells. (**B**) Two clones from LAMA-84 cells, expressing alternatively low and high PTPRG levels, were selected and screened by qRT-PCR. Additionally, immunoprecipitation, combined with Western blotting analysis, confirmed PTPRG protein expression in these clones. (**C**) Inhibition of PTPRG correlates with BCR-ABL1 tyrosine phosphorylation enhancement, and restored expression of total and phospho-β-catenin in K562 cells transfected with PTPRG FL. As expected, there was no response to the PTPRG inhibitor (two hours-treatment) in K562 transfected with the empty vector. (**D**) LAMA-84 clones expressing low or high PTPRG protein were treated with PTPRG IN for two hours. As anticipated, loss of PTPRG activity is associated with restored tyrosine phosphorylation and total β-catenin expression, only in the high expressing clone. (**E**) PTPRG-positive and negative K562 cells were treated with a specific inhibitor of BCR-ABL1, imatinib mesylate (IM), at 5 µM concentration for two hours. We observed, especially in combination with PTPRG transfection, marked dephosphorylation of β-catenin in parallel with its degradation. (**F**–**G**) Histograms representing densitometric quantification of Western blot signals from Figure 1C,D performed using ImageJ (U. S. National Institutes of Health, Bethesda, Maryland, USA, https://imagej.nih.gov/ij/, 1997–2018). Only the two high-PTPRG conditions are statistically significant. (**H**–**I**) PTPRG down-regulation by RNA interference, through a specific siRNA (siPTPRG), as assessed by qRT-PCR and Western blot, and expressed as fold of increase. PTPRG down-regulation by siRNA is associated with the up-regulation of total and pY654β-catenin. The lack of signal in the line corresponding to total β-catenin is due to the different affinity of the individual antibodies (including secondary antibodies). The result is further confirmed by immunofluorescence analysis shown in figure **(J**) both pY654 β-catenin and total β-catenin protein (red) are increased in PTPRG (green) silenced cells. The scale bar length represents 10 or 20 µm. Down-regulation of PTPRG through transfection of specific siRNA in unfractionated LAMA-84 restores the detectable quantity of phospho- and total β-catenin. Pictures are representative of at least three experiments. Fold of increase in the graphics is the mean values of 3 biological replicates. *p*-value < 0.05 was considered statistically significant. Annotations for * *p*-value < 0.05, ** *p*-value < 0.01, *** *p*-value < 0.001, and **** *p*-value < 0.0001 are provided accordingly. Error bars indicate the SD for the three replicates.

**Figure 2 ijms-21-02298-f002:**
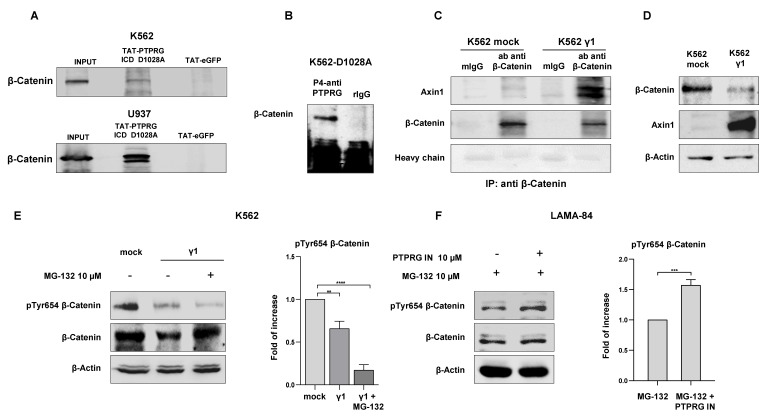
PTPRG regulates β-catenin degradation and belongs to the same protein complex. (**A**) Purified PTPRG substrate-trapping mutant, termed TAT-PTPRG ICD (intracellular domain) D1028A, includes alanine instead of aspartic acid in position 1028 that blocks phosphatase catalysis and, consequently, traps the substrate in its catalytic pocket. TAT-eGFP (enhanced Green Fluorescent Protein) was used as a negative control. β-catenin is detectable in K562 and U937 (respectively, BCR-ABL1-positive and -negative cells), indicating that β-catenin and PTPRG belong to the same protein complex, regardless of the presence of BCR-ABL1. (**B**) Co-immunoprecipitation analysis in K562 cells expressing inactive mutant PTPRG D1028A (able to bind the substrate, but not dephosphorylate it) shows that PTPRG binds β-catenin. Rabbit IgG are employed as a negative control. (**C**) Co-immunoprecipitation performed using an antibody against β-catenin, followed by Western Blotting with anti-Axin1 antibody, established the binding between these two proteins in PTPRG-positive K562 cells. (**D**) In the presence of PTPRG, Axin 1 is expressed at significantly higher levels. K562 cells transfected with PTPRG FL show a strong up-regulation of Axin1, in parallel with β-catenin down-regulation. (**E**) β-catenin proteolysis was blocked in K562 cells with an inhibitor of proteasome 26S subunit, MG-132, at the concentration of 10 µM, for two hours. Despite restored expression of total β-catenin, the Y654 phosphorylation level remains lower in the presence of PTPRG, compared with cells transfected with the empty vector. Additionally, the FOI histogram represents the ratio of phospho β-catenin to total β-catenin. Densitometric analysis were performed using ImageJ (U. S. National Institutes of Health, Bethesda, Maryland, USA, https://imagej.nih.gov/ij/, 1997–2018). Annotations for ** *p*-value < 0.01, *** *p*-value < 0.001, and **** *p*-value < 0.0001 are provided accordingly. (**F**) The combined treatment with the proteasome inhibitor MG-132 and PTPRG IN, for two hours, gives further evidence that PTPRG has a role in β-catenin tyrosine dephosphorylation in PTPRG-expressing LAMA-84: after blocking of β-catenin proteolysis, its phosphorylation level increases with the inhibition of PTPRG phosphatase activity. Pictures are representative of a minimum of three experiments.

**Figure 3 ijms-21-02298-f003:**
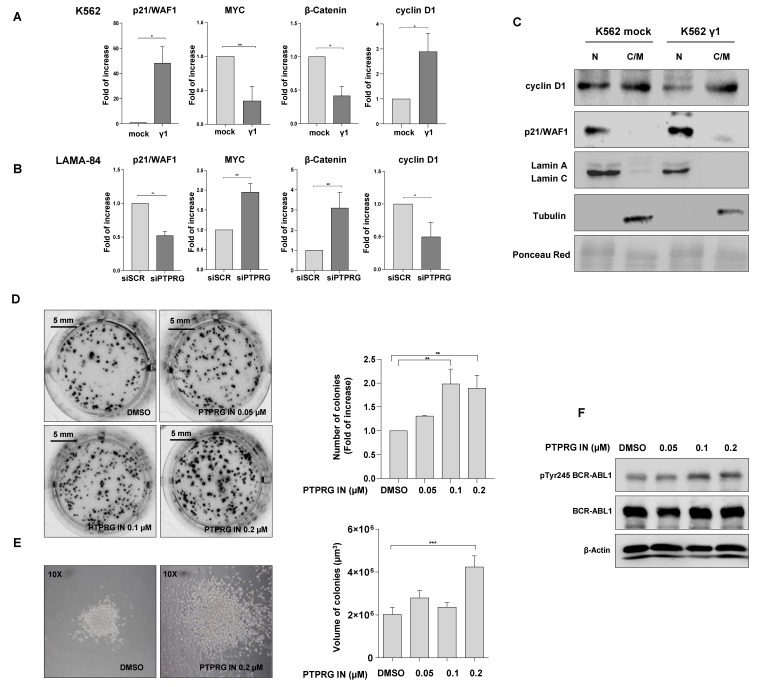
PTPRG slows down CML cell proliferation affecting β-catenin transcriptional targets. (**A** and **B**) Modulation of PTPRG expression, by transfection in K562 cells or by RNA interference in LAMA-84 cells, affects β-catenin transcriptional targets, i.e., down-regulation of *p21/WAF1* and up-regulation of *β-catenin* and *MYC*, as demonstrated by qRT-PCR experiments. Surprisingly, PTPRG expression positively correlates with the expression of *Cyclin D1* mRNA, as demonstrated by qRT-PCR made after PTPRG modulation, both in K562 and unfractionated LAMA-84 cells. (**C**) Western Blotting shows the accumulation of Cyclin D1 in the cytosolic compartment for K562 transfected with PTPRG FL, suggesting that in these cells, Cyclin D1 does not affect cell cycle progression. Moreover, the up-regulation of p21/WAF1 in the presence of PTPRG occurs both at mRNA (Figure 3A) and protein level (this panel). Lamin A/C and Tubulin proteins were used as nuclear and cytosolic markers, respectively. (**D**,**E**) Unfractionated LAMA-84 cells treated with PTPRG IN (8 days with the indicated doses) showed higher clonogenic and proliferative capability, with significant differences in size and number of colonies grown in methocult media. (**F**) Western Blotting analysis depicts the BCR-ABL1 Y245 phosphorylation level after treatment with PTPRG IN with the indicated doses. Pictures are representative of at least three experiments. Fold of Increase in the graphics is the mean values of 3 biological replicates. *p*-value < 0.05 was considered statistically significant. Annotations for * *p*-value < 0.05, ** *p*-value < 0.01, and *** *p*-value < 0.001 are provided accordingly. Error bars indicate the SD for the three replicates.

**Figure 4 ijms-21-02298-f004:**
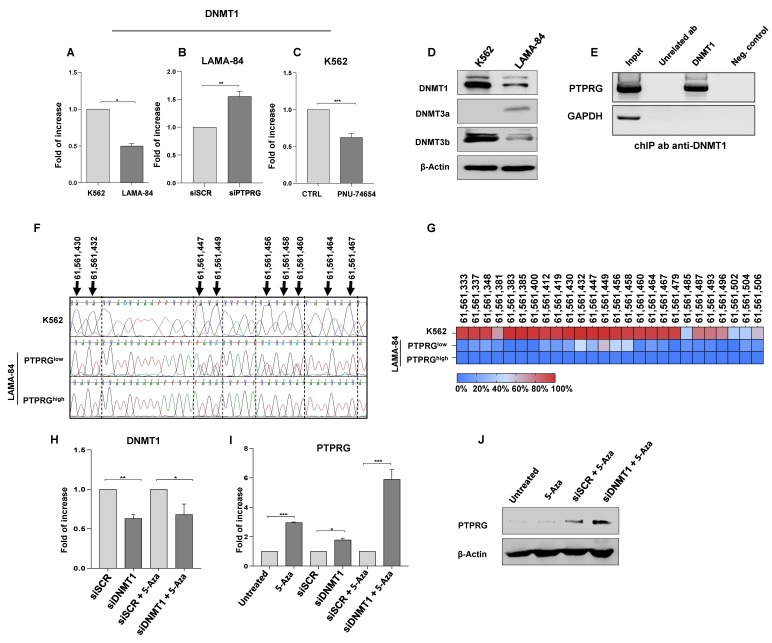
DNA (cytosine-5)-methyltransferase 1 (DNMT1) is involved in the control of PTPRG transcription. The expression of *DNMT1* mRNA correlates with β-catenin activity in CML cells. (**A**) qRT-PCR showed higher expression of *DNMT1* in K562 as compared to the LAMA-84. (**B**) Conversely, PTPRG down-regulation by a specific siRNA (siPTPRG) in LAMA-84 cells enhances *DNMT1* levels. The control is LAMA-84 transfected with a scrambled siRNA sequence (siCTRL). (**C**) The treatment with β-catenin inhibitor PNU-74654 (20 µM concentration for 12 hours) results in a significant decrease of *DNMT1* expression. (**D**) DNA-methyltransferase protein expression in CML cells: DNMT1 and DNMT3b, which cooperate for silencing a number of tumor suppressor genes in many cancer types, are up-regulated in PTPRG-negative cells. (**E**) Chromatin immunoprecipitation made in K562 cells, showing that DNMT1 binds the *PTPRG* promoter region, thus suggesting the contribution of DNMT1 to PTPRG methylation-dependent silencing. *GAPDH* amplification confirms the specificity of immunoprecipitation and PCR reactions. (**F**) Graphic representation of the methylation status of the CpG islands inside the *PTPRG* promoter region after bisulfite conversion in K562 and LAMA-84 high- and low- PTPRG clones. This technique converts unmethylated cytosines to uracil and, during PCR amplification, they are recognized as thymines (red peaks). Methylated cytosines are unaffected by bisulfite treatment, and, in the representative chromatograms (blue peaks), they are indicated by the arrows. (**G**) Heat map representing methylation pattern of specific genomic sites inside the *PTPRG* promoter region in K562, and high and low PTPRG-LAMA-84 clones. (**H**) qRT-PCR confirms the down-regulation of *DNMT1* mRNA in K562 cells after siRNA transfection, also in combination with 5-azacytidine(72 h-treatment), an inhibitor of DNA-methyltransferases, that inserts a fifth genomic base, a cytosine, with an N in position 5 of pyrimidine ring that cannot be methylated. (**I**,**J**) The DNMT1 silencing through a specific siRNA against DNMT1 as well as 5-azacytidine treatment (72 h) leads to a significative up-regulation of *PTPRG* mRNA in K562 cells. The combination of these two treatments induces a 5-fold increase of *PTPRG* mRNA expression, that becomes detectable at the protein level. Pictures are representative of at least three experiments. Fold of increase in the graphics is the mean values of 3 biological replicates. *p*-value < 0.05 was considered statistically significant. Annotations for * *p*-value < 0.05, ** *p*-value < 0.01, and *** *p*-value < 0.001 are provided accordingly. Error bars indicate the SD for the three replicates.

**Figure 5 ijms-21-02298-f005:**
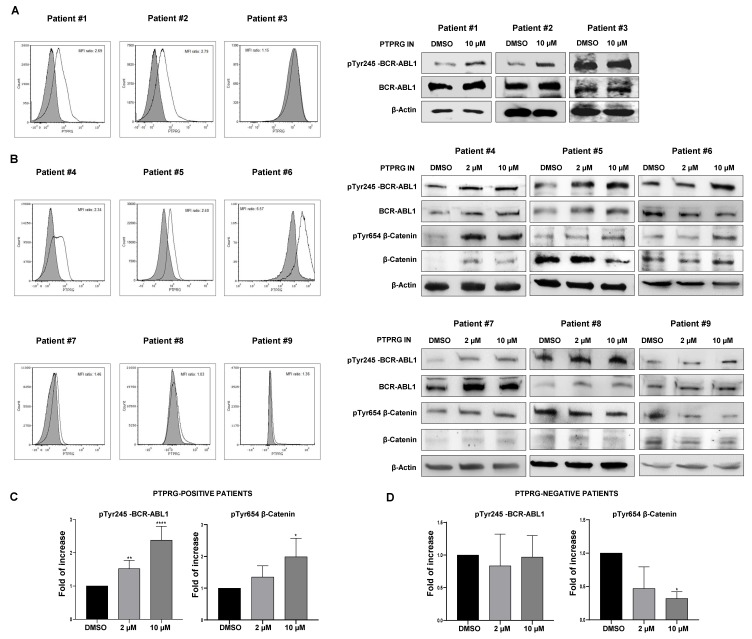
PTPRG dephosphorylates BCR-ABL1 and β-catenin in Chronic Myeloid Leukemia (CML) primary cells. (**A**,**B**) Flow cytometry analysis shows PTPRG expression in all untreated CML patients (left panels). Total peripheral leukocytes from five PTPRG-positive patients (#ID: 1, 2, 4, 5, 6) and from four lacking detectable levels of the phosphatase (#ID: 3, 7, 8, 9) were treated with 2 and/or 10 µM PTPRG IN for 2 h. Enhanced Tyr245-BCR-ABL1 phosphorylation was observed by Western Blotting analysis in cells from PTPRG-positive patients (right panel A and B), as well as Tyr654-β-catenin phosphorylation (right panel B), thus confirming that PTPRG controls the levels of BCR-ABL1 and β-catenin phosphorylation and function also in primary cells. (**C**,**D**) Densitometric analysis of Western blotting signals performed using ImageJ (U. S. National Institutes of Health, Bethesda, Maryland, USA, https://imagej.nih.gov/ij/, 1997–2018) compares the expression of pTyrBCR-ABL1 and pTyr654-β-catenin in all patients before and after the treatment with PTPRG IN. Annotations for * *p*-value < 0.05, ** *p*-value < 0.01, and **** *p*-value < 0.0001 are provided accordingly. Error bars indicate the SD for the different samples.

**Figure 6 ijms-21-02298-f006:**
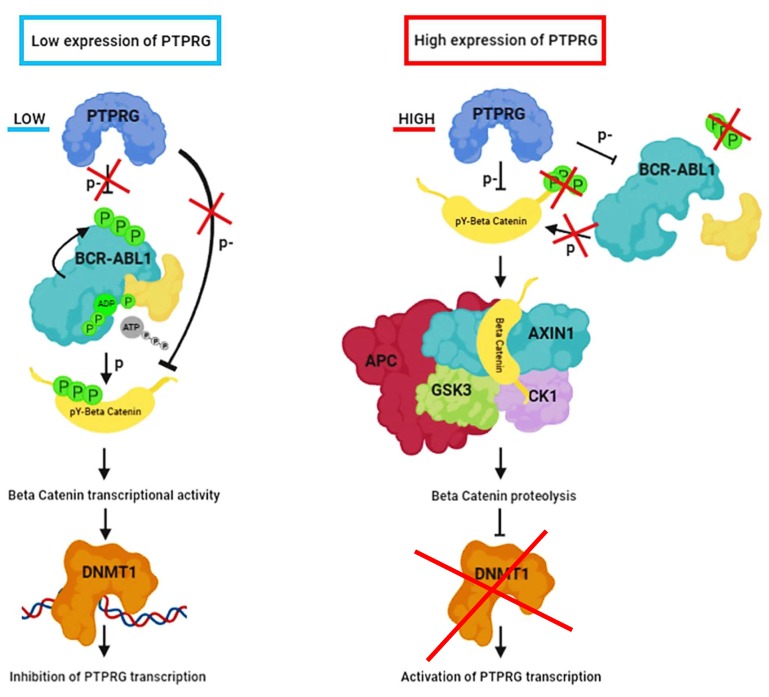
Schematic representation summarizing the regulative loop between β-catenin and PTPRG. In low-PTPRG expressing cells, BCR-ABL1 phosphorylates β-catenin in Tyrosine 86 and 654 in CML cells. This phosphorylation affects β-catenin cytosolic stabilization, blocking its binding with the destruction complex. This condition allows β-catenin to translocate into the nucleus and act as a transcriptional activator for many genes, including DNMT1 that is responsible for *PTPRG* promoter methylation and consequent down-regulation. On the other hand, in high-PTPRG expressing cells, PTPRG dephosphorylates BCR-ABL1, preventing β-catenin tyrosine phosphorylation; in addition, PTPRG directly dephosphorylates β-catenin, causing its proteolysis through the binding with its degradation complex. This condition affects the DNMT1 transcription, with the consequent down-regulation of this methyl-transferase, and hypo-methylation of *PTPRG* promoter region.

**Table 1 ijms-21-02298-t001:** Patient characteristics.

Pat.#	Age	M/F	PTPRG Expression (MFI Ratio)	Hb (g/dL)	WBC (10^9^/L)	Cytogenetics	PB Blasts (%)	BM Blasts (%)	BCR-ABL1	SOKAL Risk
**1**	32	M	2.69	10.4	360.5	46,XY,t(9;22)(q34;q11)	0.61	4	b2a2	High
**2**	75	F	2.79	10.5	186.4	46,XX,t(9;20;22)(q34;q13.1;q11)	5.32	4.9	b3a2	High
**3**	37	M	1.15	13.3	69.6	46,XY,t(9;22)(q34;q11)	0.97	1	b3a2	Interm
**4**	39	M	2.34	14.8	42.35	46,XY,t(2;9:22)(p15;q11)	0.22	0.41	b3a2	Low
**5**	45	F	2.50	12.2	16.21	46,XX,t(9;22)(q34;q11)	0.15	0.11	b3a2	Low
**6**	65	M	6.57	15.8	33.07	46,XX,t(9;22)(q34;q11)	0.2	0.28	b2a2	Low
**7**	74	M	1.46	12.7	23.13	46,XX,t(9;22)(q34;q11)	0.09	0.43	b2a2	Interm
**8**	64	M	1.03	10.2	28.36	46,XX,t(9;22)(q34;q11)	0.03	no BM	b2a2	Low
**9**	71	F	1.36	11.4	19.88	46,XX,t(9;22)(q34;q11)	0.25	0.94	b3a2	Interm

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
