# Peer review of "Regulative Loop between β-catenin and Protein Tyrosine Receptor Type γ in Chronic Myeloid Leukemia"

_ijms, 2020, doi:10.3390/ijms21072298_

Round 1
Reviewer 1 Report
The authors previously reported that PTPRG, a tyrosine phosphatase receptor type gamma, is a functional tumour suppressor, specifically down-regulated in chronic myeloid leukaemia (CML), and demonstrated its direct interaction with CML driving oncoprotein, BCR/ABL. In their current study, they associated PTPRG with potential downstream effectors, b-catenin and DNMTs and demonstrated a potential regulative loop required for BCR/ABL driven CML cell proliferation. This study is largely demonstrated in two human CML cell lines, K562 and LAMA-84 as models and at the end they went on to examine whether such regulatory circuit exists in primary CML patient blasts. This paper revealed some interesting observations, there are a number of points as follows, mainly related to the poor quality of data presentation:
1) There are a number of bar graphs with p value presented but no error bars: Figure 1F, Figures 2E/F both right panels, Figure 3F.
2) In figure 1A, please specify (I assume) it depicts the fold change of expression relative to K562 mock.
3) The resolution of all flow cytometry figures is in general poor; I can’t read any labels in figure 1F and Figures 5A/B. Please provide improved figure images, quantified graphs for their FOI comparison with appropriate statistical analysis.
4) There is no control staining of total beta-catenin images in Figure 1H.
5) Authors demonstrated a very clear isolation of two subclones from LAMA-84 with low and high expression of PTPRG shown in Figure 1A-D. It is not clear whether they continued their investigation with PTPRG high LAMA84 cells in Figures 1F-H, Figure 3B, and Figures 4 A/C/D or unfractioned cells, please clarify in text and figure legend accordingly. By using unfractioned LAMA-84, I would expect a compromised correlation between the expression of PTPRG and its downstream effectors, e.g. Figure 1G vs Figure 1D, Figure 4D.
6) I am not sure it is appropriate expression in line 137-138 stating “… belong to the same protein group”, I assume authors meant these proteins may present in the same protein complex?
7) Line 191-192 is confusing, please rephrase.
8) In Figure 3E, please specify the scale bar correlated to data shown in Figure 3G. The most straightforward data to characterise cell proliferation should be cell count/alamarblue assay in addition to CFU assay shown in Figures 3E-G.
9) Line 240-241: please add reference accordingly.
10) In general, legends in Figures 1-2 are described in different and chaotic format (e.g. A or (A)) and order (e.g. A:…. in figure 1-2 whereas in figure 3-4, legends are written as ….(A).) from Figures 3 or 4.
11) Line 292-293 and Figure 4E: ChIP experiments can only demonstrate a potential binding of DNMT1 protein to promoter region of PTPRG but not sufficient to demonstrate “a direct binding between DNMT1 and CpG islands within the promoter region.
12) Figures 5A and 5B lack WB blots for total BCR-ABL1 and beta-Catenin. There is no solid biochemistry (enzymatic activity in vitro/vivo) data shown in this study to demonstrate PTPRG directly dephosphorylates Y245 in BCR-ABL1 and Y655 in b-Catenin neither in human CML cell line nor in patient blasts (unless it has been demonstrated in the previous publications by the same author group or others? Then please provide specific supporting references). The trend of described correlation maybe evident but not being properly quantified in Figure 5 and there is no sufficient statistical analysis applied to support the claimed correlation (authors may try to quantify their WB blot data via density values?).
13) There is no data/link established in this study supporting a relationship between b-catenin and DNMT1 in their models of study, therefore part of the schematics shown in Figure 6 is not an accurate summary of this study, please revise accordingly.
In summary, the data presented in this study is of interesting finding, but appears patchy and inconsistent throughout the manuscript. With regrets, I would suggest a major revision before a decision can be made.
Reviewer 2 Report
The authors have been studying the role of the Phosphatase Receptor type γ (PTPRG), a transmembrane receptor involved in a variety of cellular processes from cell growth to oncogenic transformation. This protein has already been shown to be downregulated in CML, and when re-expressed after targeted therapies at least in some patients, can decrease BCR-ABL protein phosphorylation and thus some of its targets like CRK-L or STAT5.
They explore the link between PTPRG and β-catenin, a other known target of BCR-ABL. They have characterized a regulative loop between β-catenin and PTPRG in CML : in a low-PTPRG context, BCR-ABL directly phosphorylates β-catenin, preventing its degradation and allowing it to translocate in nucleus which down-regulates PTPRG expression through activation of DNMT1. On the other hand, when PTPRG expression is high, the late one dephosphorylates BCR-ABL so β-catenin is no longer phosphorylated and so degradated by proteolysis.
The manuscript is very well organized. The authors nicely introduced the chronic myeloid leukemia disease (CML) and their research topic. The results are very convincing and the logic behind the construction of figures is well explained.
However, I would like to make several points.
Point1: In Figure 1f, 4f, 5a and 5b the histograms are pixelized and thus unreadable.
Point2: authors should add PTPRG fold increase for better reading in Fig 1a, 1b and 1f :
Point3: authors should add anti-PTPRG band to confirm the correlation between β-catenin expression and PTPRG’s one in Fig 1c, 1d and 1e (PTPRG expression after imatinib treatment ?)
Point4: authors should add clonogenicity pictures of PTPRG IN 0.05 and 0.1 µM to make a better fit with fig 3f, 3g and 3h.
Reviewer 3 Report
In the manuscript "Regulative loop between β-Catenin and Protein Tyrosine Receptor type γ in Chronic Myeloid Leukemia", Tomasello and coworkers show for the first time that, in BCR-ABL1-related Chronic Myeloid Leukemia, the interaction between b-catenin and PTPRG activates a regulative loop controlling BCR-ABL1 chimeric onco-protein phosphorylation and activity. Given the fact that phosphorylated BCR-ABL1 promotes cell proliferation and survival, the authors demonstrated that PTPRG inhibition increases both BCR-ABL1 and b-catenin expression and phosphorylation, whereas PTPRG expression increases b-catenin degradation. The authors also demonstrated, by pull-down and immunoprecipitation experiments, that PTPRG and b-catenin form a complex regardless of the BCR-ABL1 presence. In this regulative loop, also affected is the expression of b-catenin target genes, including DNMT1. Intriguingly, the authors performed ChIP experiments and found that DNMT1 directly binds PTPRG gene promoter thereby regulating its expression. This is a well-structured paper, dealing with a topic of interest in CML research and treatment. The authors performed several experiments and appropriately discussed the results, in the context of CML knowledge. Overall the study is of significant interest for the field.
Only a few small changes need before the paper may be suitable for publication.
Minor remarks:
In the title, as well as in the first sentence of the abstract (line 25), the full name of the receptor, i.e. protein phosphatase tyrosine receptor type γ, should be reported
Line 64: change the type of parenthesis for the 12,13 references.
Figure 1C and D: To make the relationship between the level/inhibition of PTPRG and b-catenin and chimera protein amount/phosphorylation more evident, the authors should include graphs showing densitometric quantification of the WB signals.
Figure 2: As authors state that pictures are representative of at least three experiments, they must indicate the standard deviations, in the graphs E and F.
Figure 3: In panel F, authors should indicate the standard deviation of the 0.05 mM inhibitor concentration.
Figure 4. Please, invert panel B and C.
Round 2
Reviewer 1 Report
Thank authors' good effort to provide revised manuscript. I am satisfied with the rebuttal letter provided by authors and happy to endorse the manuscript to be published in present revised form.